# MOC: Multi-Order Communication in LLM-based Multi-Agent Systems

Yao Guan [1]  Lin Wang [1]  Zhihu Lu [1]  Ziyi Wang [1]  Wenzhu Yan [2]  Qiang Duan [3]

## Abstract

Despite the remarkable progress of Large Language Model (LLM) based Multi-Agent Systems, most research focuses on optimizing coordination topology while largely underexploring the equally critical problem: *how to transmit and optimize messages among agents effectively?* Current communication schemes typically rely on the direct concatenation of first-order neighbor responses, which induces a restricted evidence receptive field and leads to the dilution of crucial insights over multi-hop paths. To address these limitations, we propose the Multi-Order Communication (MOC) scheme, which reconstructs the inter-agent communication to capture multi-hop dependencies and incorporates a structural message consolidation strategy to ensure efficiency. Specifically, we formalize the communication mechanism to construct a structured multi-order evidence stream, and subsequently design a Semantic-Topological Merging algorithm to optimize semantic fidelity within token constraints. Extensive experiments across six diverse datasets and LLM backbones of varying parameter scales demonstrate that MOC consistently improves task performance and reduces communication costs. The code is available at
`https://github.com/yao-guan/MOC`.

## 1. Introduction

Large Language Model (LLM) (Zhao et al., 2023; Chang et al., 2024; Naveed et al., 2025) based Multi-Agent Systems (MAS) have emerged as a promising paradigm for complex problem-solving tasks that require long-horizon reasoning, tool use, and reliable verification (Guo et al., 2024; He et al., 2025). Through orchestrating collectives of

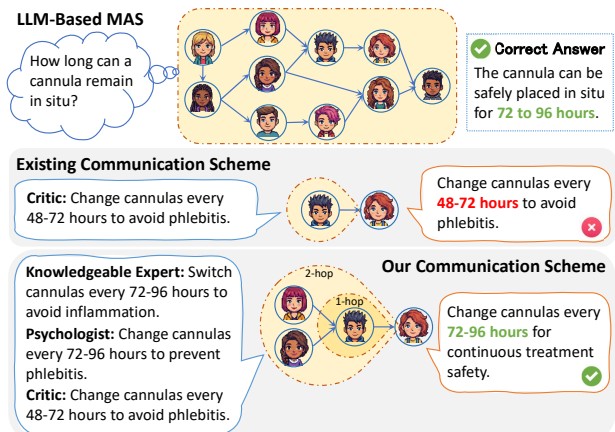

*Figure 1.* The paradigm comparison between existing communication scheme and ours.

autonomous agents to engage in multi-step planning and iterative deliberation, MAS can effectively decompose intricate objectives into manageable sub-goals, assign specialized roles to distinct agents, and facilitate parallel exploration of the solution space (Li et al., 2023; Hong et al., 2024; Wu et al., 2024; Tran et al., 2025). Such emergent collective intelligence is fundamentally governed by the optimization of multi-agent topology and communication scheme, i.e., how agents are connected and how they communicate.

Recent works have predominantly focused on topology design for multi-agent coordination (Hong et al., 2024; Zhuge et al., 2024; Liu et al., 2024b; Wu et al., 2024; Zhang et al., 2025b; Qian et al., 2025), exploring how to structure agent connectivity to enable effective collaboration. A broad spectrum of topologies has been explored, ranging from lightweight structures such as chain (Hong et al., 2024) and tree (Wu et al., 2024), to dense or stochastic connectivity such as fully connected and random graphs (Qian et al., 2025), and more recently to task-adaptive dynamic topologies that generate or reconfigure connectivity conditioned on the task (Zhuge et al., 2024; Liu et al., 2024b; Zhang et al., 2025b). While these topology-centric studies verify that appropriately structuring agent interactions can improve coordination under limited supervision, an equally critical yet overlooked question remains: ***Beyond connecting agents, how should we structure the communication scheme to transmit and optimize messages effectively?***

[1]Fudan University  [2]Nanjing Normal University  [3]Pennsylvania State University. Correspondence to: Zhihu Lu <lzh@fudan.edu.cn>.

*Proceedings of the 43rd International Conference on Machine Learning*, Seoul, South Korea. PMLR 306, 2026. Copyright 2026 by the author(s).

In practice, the underlying communication mechanism remains rudimentary (Chen et al., 2024a), as existing LLM-based MAS mostly adopt a simple communication scheme that directly concatenates outputs from immediately connected predecessor agents. This rigid interaction paradigm suffers from critical limitations stemming from **Restricted Evidence Receptive Field**. As illustrated in Figure 1, such systems typically restrict access to direct upstream neighbors, inducing a narrow first-order scope. Crucially, this restriction forces rich, multi-hop contexts to be progressively homogenized into immediate predecessors' outputs, failing to preserve diverse insights. Consequently, the system underutilizes its collaborative potential, resulting in suboptimal performance. A naive remedy is to expose the target agent to more upstream messages. However, this quickly runs into a second bottleneck of **Context Inefficiency**. Since LLMs consume ordered text rather than fixed-size vectors, expanding the evidence scope leads to rapid growth in prompt length and degraded utilization of long contexts (Jiang et al., 2024; Liu et al., 2024a).

To address these challenges, we propose **Multi-Order Communication (MOC)**, a topology-aware communication scheme that exposes a target agent to raw upstream responses from multiple hop distances within a single intra-round execution. Inspired by multi-hop aggregation in graph learning (e.g., MixHop (Abu-El-Haija et al., 2019)), MOC constructs a structured multi-order evidence stream by tracing upstream dependencies and grouping messages by hop order, enabling the target agent to reason over long-range evidence while retaining access to original sources for verification. To make such multi-order exposure practical under limited context budgets, we further introduce a structural message consolidation operator that removes redundancy before messages are concatenated into the target agent's input, while preserving the topological precedence implied by the MAS execution order. This consolidation is instantiated via semantic-topological merging, where redundant messages are identified using lightweight embeddings and the merged content is distilled with a lightweight model under explicit length control to stabilize the resulting context.

Our main contributions are highlighted below:

- **Theoretical Formalization.** We study the under-explored communication-scheme design problem in LLM-based MAS and provide a graph-based formalization of intra-round multi-agent communication, including an acyclic execution scheme and context composition for ordered textual message passing.

- **Paradigm Innovation.** We propose Multi-Order Communication (MOC), a topology-aware scheme that surfaces raw upstream evidence at multiple hop orders to the target agent in a structured prompt, enabling

long-range reasoning and source verification within a round.

- **Practical Solution.** We introduce a structural message consolidation operator to compress multi-order evidence under context budgets while preserving topological precedence, and instantiate it with a semantic-topological merging strategy.

- **Experimental Validation.** Extensive evaluations demonstrate that MOC is: (1) **high-performing**, yielding a 6.77% accuracy boost on AQuA for Gemma-2-27B, (2) **robustly scalable**, improving HumanEval by 3.68% for Qwen2.5-32B and MMLU-Pro by 0.82% for DeepSeek-V3.2 with $K=2$ identified as the optimal receptive field, and (3) **context-efficient**, as the operator $\Phi$ reduces input tokens from $13.69 \times 10^5$ to $12.49 \times 10^5$ under the 20-agent setting, which is lower than the $13.38 \times 10^5$ tokens used by Vanilla MAS.

## 2. Related Work

**LLM-based Multi-Agent Systems.** The robust foundation established by single agents (Yao et al., 2023; Crispino et al., 2024; Wu et al., 2025) has paved the way for Multi-Agent Systems (MAS), where organizing multiple LLMs into collaborative groups significantly amplifies individual model capabilities and fosters emergent behaviors (Guo et al., 2024; Li et al., 2024; Xi et al., 2025). To harness this collaborative power, recent studies have proposed diverse frameworks. For instance, CAMEL (Li et al., 2023) introduced communicative agents for autonomous cooperation via role-playing. AutoGen (Wu et al., 2024) generalized this paradigm by enabling the construction of complex applications with conversable agents through multi-agent dialogues. Similarly, LLM-Debate (Du et al., 2024) leverages a multi-agent argumentation mechanism to effectively rectify hallucinations. Furthermore, frameworks like Generative Agents (Park et al., 2023) and AgentVerse (Chen et al., 2024b) facilitate flexible collaboration by simulating dynamic social environments or orchestrating expert groups to solve complex, multi-faceted problems. Despite these advancements, the escalating complexity of agent interactions has highlighted a critical need for formal and structured architectures to govern the underlying coordination patterns.

**Graph Modeling for LLM-based MAS.** To enjoy the natural advantage of graph structures in modeling such complex inter-agent relationships (Niu et al., 2021; Hu et al., 2024; Bei et al., 2025) and the advanced reasoning and planning capabilities of LLM (Schaeffer et al., 2023; Zhao et al., 2023), researchers have increasingly adopted graph-based architectures to orchestrate LLM-based multi-agent collaboration. While early works like CAMEL (Li et al., 2023), MetaGPT (Hong et al., 2024), and AutoGen (Wu et al.,

2024) implicitly utilized graph-like patterns for coordination, subsequent frameworks such as GPTswarm (Zhuge et al., 2024), DyLAN (Liu et al., 2024b), G-Designer (Zhang et al., 2025b), and MACNET (Qian et al., 2025) explicitly cast agents as nodes and communication channels as edges in directed graphs, optimizing topologies to enhance collaboration. Building on this graph-based view, recent works optimize the interaction graph by sparsifying redundant edges (Zhang et al., 2025a) or nodes (Wang et al., 2025b) for efficiency, while employing topological interventions (Wang et al., 2025a) to block malicious propagation. However, existing research still primarily focuses on topology design, largely overlooking the communication pipeline: how semantic messages are transmitted and optimized along these edges. In this paper, we address this gap by developing an explicit message passing mechanism for inter-agent communication.

## 3. Preliminaries

In this section, we model the LLM-based MAS as a directed graph and formalize an acyclic execution scheme for inter-agent communication.

### 3.1. LLM-based MAS as Graph

We define the LLM-based multi-agent communication topology as a directed graph $\mathcal{G} = (\mathcal{V}, \mathcal{E})$, where $\mathcal{V}$ and $\mathcal{E}$ denote the sets of nodes and edges, respectively. Each node $v_i \in \mathcal{V}$ represents an LLM-based autonomous agent, which is associated with a generative function $f_i(\cdot)$ parameterized by an internal state tuple $\theta_i$:

$$\theta_i = (\text{Role}_i, \text{Plugins}_i), \qquad (1)$$

where $\text{Role}_i$ is the agent's pre-assigned role or function, and $\text{Plugins}_i$ contains external tools such as search engines and document parsers. An edge $e_{i \to j} \in \mathcal{E}$ indicates information flow from $v_i$ to $v_j$, represented by the adjacency matrix $\mathbf{A} \in \{0, 1\}^{N \times N}$, where $\mathbf{A}_{ij} = 1$ if an edge exists, and $0$ otherwise. The set of in-neighbors for agent $v_j$ is:

$$\mathcal{N}_j^{\text{in}} = \{v_i \in \mathcal{V} \mid \mathbf{A}_{ij} = 1\}, \qquad (2)$$

### 3.2. Communication Scheme on the Graph

Given a query $\mathcal{Q}$, an LLM-based MAS can be executed for multiple rounds to derive a final solution. In this work, we focus on a *single intra-round* execution at round $t$, where each agent produces one response following a topological order. For brevity, we omit the superscript $^{(t)}$ when it causes no ambiguity. Let a topological sort $\phi$ map $\mathcal{G}$ to an execution sequence $\sigma$:

$$\begin{aligned} \phi : \mathcal{G} &\longmapsto \sigma, \quad \sigma = [v_{\sigma_1}, v_{\sigma_2}, \ldots, v_{\sigma_N}], \\ s.t. \quad &\forall 1 \le p < q \le N, \ v_{\sigma_q} \notin \mathcal{N}_{\sigma_p}^{\text{in}}, \end{aligned} \qquad (3)$$

This acyclic execution order ensures agents execute only after their dependencies are met. Therefore, we restrict the intra-round communication topology $\mathcal{G}$ to be a directed acyclic graph (DAG), a common choice in orchestrated LLM-based MAS pipelines (Hong et al., 2024; Zhuge et al., 2024; Zhang et al., 2025b; Wang et al., 2025b). Specifically, the communication logic for node $v_j$ is characterized by a two-stage process of aggregating discrete responses from $\mathcal{N}_j^{\text{in}}$ into a synthesized context $\mathcal{C}_j$ followed by conditioned response generation. Analogous to the Message Passing (MP) rule in GNNs (Kipf & Welling, 2017; Li et al., 2018; Wu et al., 2019), we formulate the communication scheme via two key operations: *In-neighbor Context Composition* and *Agent Response Generation*. A detailed comparison with standard MP is deferred to the appendix A.

**Definition 3.1. (In-neighbor Context Composition.)** Let $v_j$ be an LLM-based agent with in-neighbor set $\mathcal{N}_j^{\text{in}}$. In the considered intra-round execution, the discrete natural language responses $\{\mathcal{R}_i\}_{v_i \in \mathcal{N}_j^{\text{in}}}$ are generated following the execution order $\sigma$. The In-neighbor Context Composition operation $\Psi(\cdot)$ assembles these responses into a unified linear context $\mathcal{C}_j$. To ensure a deterministic structure, we perform ordered concatenation:

$$\mathcal{C}_j = \Psi\left(\{\mathcal{R}_i\}_{v_i \in \mathcal{N}_j^{\text{in}}} \mid \pi_j\right) = \bigoplus_{k=1}^{n_j} \mathcal{R}_{\pi_j(k)}, \qquad (4)$$

where $\bigoplus$ denotes repeated prompt concatenation over the ordered list, $n_j = |\mathcal{N}_j^{\text{in}}|$ denotes the number of in-neighbors, and $\pi_j$ is an order mapping function that sorts the neighbors. By default, $\pi_j$ is the order induced by $\sigma$ restricted to $\mathcal{N}_j^{\text{in}}$. For agents with $\mathcal{N}_j^{\text{in}} = \emptyset$, we define $\Psi(\emptyset) = \emptyset$.

**Definition 3.2. (Agent Response Generation.)** Let $v_j$ be an LLM-based agent with system prompt $\mathcal{P}_{sys,j}$ derived from its internal state $\theta_j$ (e.g., encoding its Role and available Plugins). Given a task query $\mathcal{Q}$ and the composited in-neighbor context $\mathcal{C}_j$ from Definition 3.1, the Agent Response Generation operation queries $v_j$ with $\mathcal{P}_{sys,j}$, $\mathcal{Q}$, and $\mathcal{C}_j$, then produces the natural language response $\mathcal{R}_j$:

$$\mathcal{R}_j = f_j\left(\mathcal{P}_{sys,j} \oplus \mathcal{Q} \oplus \mathcal{C}_j\right), \qquad (5)$$

where $\oplus$ denotes prompt concatenation. Specifically, when $\mathcal{N}_j^{\text{in}} = \emptyset$, we set $\mathcal{C}_j = \emptyset$, and the agent generates $\mathcal{R}_j$ conditioned only on $\mathcal{P}_{sys,j}$ and $\mathcal{Q}$.

## 4. Methodology

This section presents the Multi-Order Communication (MOC) scheme, structured into two components: the **Proposed Communication Scheme** (Section 4.1), which defines the topological tracing of multi-hop dependencies and highlights the necessity of handling context efficiency, and **Optimizing Message Consolidation** (Section 4.2), which

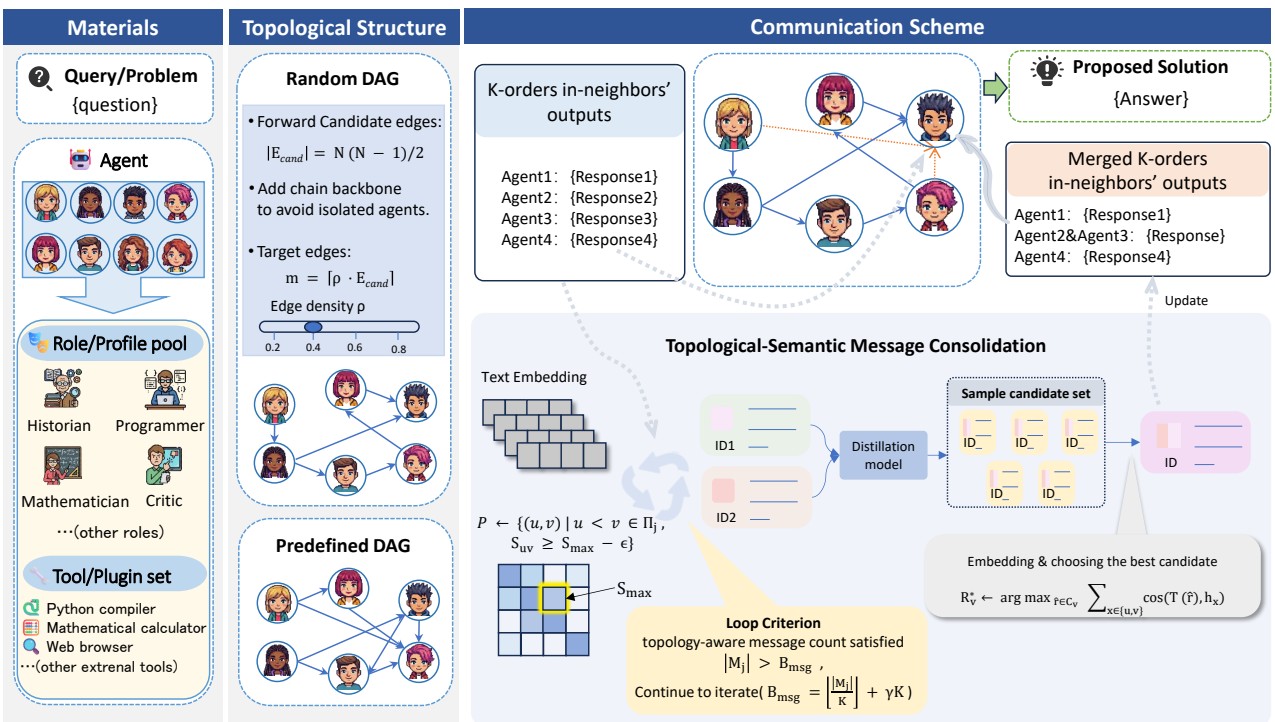

*Figure 2.* The overview of our proposed multi-order communication scheme.

instantiates the consolidation operator $\Phi$ to balance semantic fidelity with token constraints. Figure 2 illustrates the MOC framework.

### 4.1. Proposed Communication Scheme

**Theoretical Inspiration.** In the naive LLM-based MAS communication scheme, the target agent acquires context solely from its direct upstream neighbors. Consequently, intermediate agents serve as the exclusive conduits for distant information, where their paraphrasing, summarization, and reinterpretation of upstream outputs inevitably introduce semantic attenuation (Zhou et al., 2024; Zhang et al., 2024a), leading to accumulated omissions and biases across hops. Motivated by MixHop (Abu-El-Haija et al., 2019), which concatenates node representations aggregated from varying hop distances to expand a node's receptive field and capture long-range dependencies, we propose Multi-Order Communication (MOC) for LLM-based MAS. MOC directly exposes a target agent to raw upstream responses at multiple hop distances and organizes them into a single structured prompt following their topological dependencies to align the input with the reasoning flow. This multi-order evidence presentation expands evidence coverage within one interaction round, supports reasoning over long-range dependencies, and enables the verification of intermediate agents' restatements against original sources.

**$K$-order Message Flow Construction.** Following the intra-round topological execution sequence $\sigma$, all upstream responses that precede $v_j$ in $\sigma$ are available when executing $v_j$, allowing us to collect raw evidence from multi-hop ancestors. For a target agent $v_j$, we construct its input context by collecting upstream evidence across $K$ hop orders. We first characterize $k$-hop reachability using powers of the adjacency matrix $\mathbf{A}$. For each upstream agent $v_i$, we define its shortest reachable order to $v_j$ as:

$$k_{\min}(i,j) := \min\{k \in [1, K] \mid [\mathbf{A}^k]_{ij} > 0\}, \quad (6)$$

where $[\mathbf{A}^k]_{ij} > 0$ indicates the existence of a directed path of length $k$ from $v_i$ to $v_j$. We then assign each message to its shortest order and define the $k$-th order response set:

$$\mathcal{S}_j^{(k)} := \{\mathcal{R}_i \mid k_{\min}(i,j) = k\}, \quad (7)$$

where $k = 1, \ldots, K$. For the base case $k = 1$, $\mathcal{S}_j^{(1)}$ consists of responses from the direct in-neighbors, i.e., $\{\mathcal{R}_i \mid v_i \in \mathcal{N}_j^{\text{in}}\}$. To obtain a deterministic linearization, we order each hop set by the global topological order $\sigma$ and define

$$\pi_j^{(k)} := \text{sort}_\sigma\big(\mathcal{S}_j^{(k)}\big), \quad (8)$$

where $\text{sort}_\sigma(\cdot)$ sorts elements by their positions in $\sigma$ (restricted to the set). Finally, since each message is assigned to its shortest reachable order, the sets $\{\mathcal{S}_j^{(k)}\}_{k=1}^K$ are disjoint, and we define a far-to-near global ordering by concatenating them as follows:

$$\mathcal{M}_j := \bigcup_{k=1}^{K} \mathcal{S}_j^{(k)}, \qquad \Pi_j := \pi_j^{(K)} \parallel \pi_j^{(K-1)} \parallel \cdots \parallel \pi_j^{(1)},$$
$$(9)$$

where $\parallel$ denotes sequence concatenation. We use $\Pi_j$ to denote the resulting ordering and its induced rank function. Specifically, $\Pi_j(i)$ returns the position of message $\mathcal{R}_i$ in this ordering.

**Context-Efficiency Gap & Consolidation.** Unlike GNNs, which aggregate fixed-dimension vectors via permutation-invariant summation (Kipf & Welling, 2017; Veličković et al., 2018; Hamilton et al., 2017), *message aggregation* in LLM-based agents relies on ordered textual concatenation, causing context length to expand linearly with evidence scope. Within this sequential paradigm, the relative positioning of messages governs the agent's attention and reasoning trajectory, elevating structural coherence to a critical constraint alongside token efficiency. While the raw $K$-order message set $\mathcal{M}_j$ in Eq. (9) is theoretically comprehensive, enabling agents to synthesize richer context for improved reasoning, directly compositing these messages via $\Psi$ in Definition 3.1 may encounter efficiency bottlenecks (Jiang et al., 2024) and potential performance saturation (Shi et al., 2023; Liu et al., 2024a). To bridge this gap, we introduce a message-level consolidation operator $\Phi$ before context composition.

**Definition 4.1. Structural Message Consolidation.** Given a raw message set $\mathcal{M}_j$ and its associated ordering $\Pi_j$, the message-level consolidation operator $\Phi$ generates a consolidated tuple $(\mathcal{M}_{j,*}, \Pi_{j,*})$ that preserves semantic fidelity while minimizing the token cost for agent inference:

$$(\mathcal{M}_{j,*}, \Pi_{j,*}) = \Phi(\mathcal{M}_j, \Pi_j), \qquad (10)$$

Formally, the objective of $\Phi$ is to maximize the semantic coverage of the context subject to a predefined context budget, while preserving the message order induced by $\Pi_j$.

**Multi-Order Communication Scheme.** The proposed MOC sequentially orchestrates three defined steps: structural message consolidation $\Phi$ (Definition 4.1), context composition $\Psi$ (Definition 3.1), and the final agent response generation (Definition 3.2). Synthesizing these components, we express MOC via a unified formulation that encapsulates the entire execution workflow as follows:

$$\mathcal{R}_j = f_j\big(\mathcal{P}_{sys,j} \oplus \mathcal{Q} \oplus \big(\Psi \circ \Phi(\mathcal{M}_j \mid \Pi_j)\big)\big), \qquad (11)$$

This composite formulation ensures the context is both semantically rich and structurally efficient. Notably, in the specific case where $\Phi$ is an identity mapping and $K = 1$, MOC reduces to the conventional communication scheme, highlighting its nature as a generalized framework for multi-agent interaction.

## 4.2. Optimizing Message Consolidation

**Semantic-Topological Merging.** To bridge the gap between multi-order evidence and context efficiency, we propose a semantic-topological merging strategy designed to distill redundant information before it is concatenated into the target agent's input context, while strictly honoring the established communication hierarchy. This process begins by projecting raw textual responses into a high-dimensional semantic space to enable quantitative similarity analysis. For each message $\mathcal{R}_i$ in the multi-order set $\mathcal{M}_j$, we first compute its vector representation $\mathbf{h}_i$ using a text embedding function $\mathcal{T} : \mathbb{T} \to \mathbb{R}^d$:

$$\mathbf{h}_i := \mathcal{T}(\mathcal{R}_i), \qquad (12)$$

where $\mathbb{T}$ denotes the space of textual utterances, $d$ is the embedding dimension, and $\mathcal{T}(\cdot)$ is instantiated via a lightweight text encoder such as (Reimers & Gurevych, 2019; Wang et al., 2020).

Upon obtaining these embeddings, we compute an ordered similarity matrix $\mathbf{S} \in \mathbb{R}^{|\mathcal{M}_j| \times |\mathcal{M}_j|}$, where messages are re-indexed by their ranks induced by $\Pi_j$ and each entry $\mathbf{S}_{uv} = \cos(\mathbf{h}_u, \mathbf{h}_v)$ represents the cosine similarity between the embeddings of the $u$-th and $v$-th ranked messages. This alignment ensures that our analysis of semantic redundancy is inherently grounded in the graph's reachability and the predefined execution order. To mitigate semantic drift and resolve redundancy without disrupting the logical reasoning flow, we implement a forward-merging strategy that anchors each merge at the later message position, thereby preserving the message order induced by $\Pi_j$. We identify high-similarity pairs $(u, v)$ with $u < v$ and treat the later message $v$ as the anchor, then absorb the predecessor $u$ into it as follows:

$$\tilde{\mathcal{R}}_v = \mathcal{D}\big(\mathcal{R}_u \oplus \mathcal{R}_v\big), \qquad (13)$$

where $\mathcal{D}(\cdot)$ denotes a small pre-trained language model (e.g., (Zhang et al., 2024b; Abdin et al., 2024)) that distills the joint information. Following this synthesis, the redundant evidence is pruned to maintain structural efficiency:

$$\mathcal{M}_j \leftarrow \mathcal{M}_j \setminus \{\mathcal{R}_u\}, \quad \Pi_j \leftarrow \mathrm{Rank}(\Pi_j \setminus \{u\}), \quad (14)$$

where $\mathrm{Rank}(\Pi_j \setminus \{u\})$ updates the rank function after removing element $u$.

**Optimization Strategy.** To mitigate semantic drift and minimize information loss during distillation, we generate a candidate set $\mathcal{C}_v$ through diverse sampling of the operator $\mathcal{D}(\cdot)$, and select the optimal representation $\mathcal{R}_v^*$ that maximizes semantic proximity to its original sources:

$$\mathcal{R}_v^* = \arg\max_{\hat{r} \in \mathcal{C}_v} \sum_{x \in \{u,v\}} \cos\big(\mathcal{T}(\hat{r}), \mathbf{h}_x\big). \qquad (15)$$

We additionally impose a hard length constraint in the distillation prompt, requiring $Token(\mathcal{R}_v^*) \leq \kappa\big(Token(\mathcal{R}_u) +$

---

**Algorithm 1** Topological-Semantic Message Consolidation

---

**Input:** Raw message set $\mathcal{M}_j$, Rank function $\Pi_j$, Distillation model $\mathcal{D}$, Budget $B_{\text{msg}}$, Threshold $\epsilon$
**Output:** Consolidated tuple $(\mathcal{M}_{j,*}, \Pi_{j,*})$
**while** $|\mathcal{M}_j| > B_{\text{msg}}$ **do**
  Re-index messages by ranks induced by $\Pi_j$, denote the $u$-th ranked message and embedding as $(\mathcal{R}_u, \mathbf{h}_u)$
  Compute embeddings $\mathbf{h}_u \leftarrow \mathcal{T}(\mathcal{R}_u)$ for all ranked messages $\mathcal{R}_u \in \mathcal{M}_j$
  Obtain similarity matrix $\mathbf{S}$ where $\mathbf{S}_{uv} \leftarrow \cos(\mathbf{h}_u, \mathbf{h}_v)$
  Identify candidate pairs $\mathcal{P} \leftarrow \{(u,v) \mid u < v, \mathbf{S}_{uv} \geq \mathbf{S}_{\text{max}} - \epsilon\}$
  $\mathcal{P} \leftarrow$ maximal disjoint set of pairs extracted from $\mathcal{P}$
  **if** $\mathcal{P}$ is empty **then**
    **break** {No redundant pairs exceed the threshold}
  **end if**
  **for each** $(u,v) \in \mathcal{P}$ **do**
    Sample candidate set $\mathcal{C}_v$ via $\mathcal{D}(\mathcal{R}_u \oplus \mathcal{R}_v)$
    $\mathcal{R}_v^* \leftarrow \arg\max_{\hat{r} \in \mathcal{C}_v} \sum_{x \in \{u,v\}} \cos(\mathcal{T}(\hat{r}), \mathbf{h}_x)$
    Update content: $\mathcal{R}_v \leftarrow \mathcal{R}_v^*$
    Prune message set: $\mathcal{M}_j \leftarrow \mathcal{M}_j \setminus \{\mathcal{R}_u\}$
    Update rank function: $\Pi_j \leftarrow \text{Rank}(\Pi_j \setminus \{u\})$
  **end for**
**end while**

---

$Token(\mathcal{R}_v))$ with $\kappa < 0.5$, where $Token(\cdot)$ is estimated using the target LLM tokenizer. The distillation prompts are provided in Appendix B.3. In practice, to handle variable per-message lengths, we operationalize the context budget in Definition 4.1 using a *topology-aware message-count* constraint $B_{\text{msg}}$ as the termination criterion:

$$B_{\text{msg}} = \left\lfloor \frac{|\mathcal{M}_j|}{K} \right\rfloor + \gamma K. \tag{16}$$

where $\gamma$ denotes a hyperparameter for long-range evidence tolerance. We compute $B_{\text{msg}}$ once from the initial message set $\mathcal{M}_j$ and keep it fixed throughout consolidation. The consolidation process iterates until the condition $|\mathcal{M}_j^*| \leq B_{\text{msg}}$ is satisfied, effectively stabilizing the signal-to-noise ratio within the multi-order flow. Finally, to accelerate this reduction process, we implement a *batch-wise $\epsilon$-approximate merging strategy*. Rather than sequential greedy updates, we simultaneously merge a maximal disjoint set of pairs satisfying $\mathbf{S}_{uv} \geq \mathbf{S}_{\text{max}} - \epsilon$ in each iteration. This parallelization enables a logarithmic reduction in consolidation rounds, significantly decreasing computational latency while strictly maintaining the topological constraints and the structural coherence of the consolidated stream. The complete optimization procedure is formalized in Algorithm 1.

**Complexity Analysis & Discussion.** The MOC framework improves computational efficiency through structural decoupling and selective information distillation, characterized by the following properties:

(1) *Bounded Target-Agent Attention.* By enforcing the message-count budget $B_{\text{msg}}$, the target agent attends over at most $B_{\text{msg}}$ messages, yielding a message-level attention proxy of $\mathcal{O}(B_{\text{msg}}^2)$ versus $\mathcal{O}(|\mathcal{M}|^2)$ without consolidation. With the imposed length cap on distilled outputs, the target-agent context length is effectively controlled in practice. We report token usage in experiments.

(2) *Batch-wise Latency Reduction.* In each iteration, we merge a maximal disjoint set of high-similarity pairs. This batch-wise merging can reduce the number of consolidation rounds in practice, and it admits a best-case $\mathcal{O}(\log |\mathcal{M}|)$ behavior under favorable disjoint-pair structures. This improves practical efficiency in dense agent networks.

(3) *Similarity-Driven Adaptation.* By prioritizing highly similar message pairs early, the framework quickly removes semantic redundancy across agent responses and reduces the effective message count. This allows the remaining budget to focus on more diverse evidence and higher-order dependencies under fixed resources.

## 5. Experiments

### 5.1. Experimental Setup

**Models & Datasets.** We conduct experiments using models of varying scales, including Gemma-2-27B-Instruct (Team, 2024), Qwen2.5-32B-Instruct (Qwen et al., 2025), and DeepSeek-V3.2-685B-Chat (DeepSeek-AI et al., 2025). We evaluate the effectiveness of MOC across six datasets spanning three representative categories: (1) General Reasoning: MMLU (Hendrycks et al., 2021), MMLU-Pro (Wang et al., 2024); (2) Mathematical Reasoning: GSM8K (Cobbe et al., 2021), SVAMP (Patel et al., 2021), AQuA (Ling et al., 2017); (3) Code Generation: HumanEval (Chen et al., 2021). Detailed dataset statistics are in AppendixB.1.

**Topological Backbones.** For multi-agent communication topologies, we adopt a randomly constructed DAG backbone, where we sample a random topological order and only allow forward edges to ensure acyclicity while avoiding isolated agents, Appendix B.2 shows the exact construction and edge density control.

**Implementation Details** For experiments with Gemma-2-27B-Instruct and Qwen2.5-32B-Instruct, we perform inference using Ollama on Nvidia RTX-4090 GPUs and set the temperature to 0. For experiments with DeepSeek-V3.2-685B-Chat, we access the model via APIs and set the temperature to 1. The text encoder is implemented using (Wang et al., 2020) with the embedding dimension set to D = 384. The distillation model is gemma2-9B-Instruct (Team, 2024), sampling times M are set as 5, and $\kappa$=0.45. The hyperparameter $K = 2$, $\gamma = 1$, and $\epsilon = 0.1$.

*Table 1.* Main evaluation results of **seven** *Gemma-2-27B-Instruct* based agents across various benchmarks under different edge densities $\rho$. '+' denotes the integration of MOC. Improv. indicates the relative percentage improvement achieved by MOC compared to the Vanilla MAS baseline. Bold values represent the best performance.

| Edge Density | Method | MMLU | MMLU-Pro | AQuA | GSM8K | SVAMP | HumanEval | Avg. |
|---|---|---|---|---|---|---|---|---|
| | Single LLM | 0.7439 | 0.4571 | 0.6535 | 0.8900 | 0.8633 | 0.7317 | 0.7233 |
| $\rho = 0.3$ | Vanilla MAS | 0.7895 | 0.5679 | 0.6969 | 0.9100 | 0.9067 | 0.7683 | 0.7732 |
| | + MOC | **0.7930** | 0.5750 | 0.7441 | **0.9300** | **0.9233** | 0.7805 | **0.7910** |
| | *Improv.* | ↑ 0.44% | ↑ 1.25% | ↑ 6.77% | ↑ 2.20% | ↑ 1.83% | ↑ 1.59% | ↑ 2.30% |
| $\rho = 0.5$ | Vanilla MAS | 0.7789 | 0.5786 | 0.7244 | 0.9167 | 0.9033 | 0.7744 | 0.7794 |
| | + MOC | 0.7895 | **0.5929** | 0.7323 | **0.9300** | 0.9067 | **0.7866** | 0.7897 |
| | *Improv.* | ↑ 1.36% | ↑ 2.47% | ↑ 1.09% | ↑ 1.45% | ↑ 0.38% | ↑ 1.58% | ↑ 1.32% |
| $\rho = 0.7$ | Vanilla MAS | 0.7860 | 0.5536 | 0.7087 | 0.9067 | 0.9067 | 0.7744 | 0.7727 |
| | + MOC | 0.7895 | 0.5607 | 0.7205 | 0.9133 | 0.9133 | 0.7805 | 0.7796 |
| | *Improv.* | ↑ 0.45% | ↑ 1.28% | ↑ 1.67% | ↑ 0.73% | ↑ 0.73% | ↑ 0.79% | ↑ 0.89% |
| $\rho = 1.0$ | Vanilla MAS | 0.7719 | 0.5643 | 0.7087 | 0.9133 | 0.9200 | 0.7622 | 0.7734 |
| | + MOC | 0.7860 | 0.5821 | 0.7283 | 0.9133 | 0.9167 | 0.7683 | 0.7825 |
| | *Improv.* | ↑ 1.83% | ↑ 3.15% | ↑ 2.77% | ↑ 0.00% | ↓ 0.36% | ↑ 0.80% | ↑ 1.18% |

*Table 2.* Evaluation results of **seven** *Qwen2.5-32B-Instruct* based agents on MMLU and HumanEval datasets across different edge densities $\rho$.

| Edge Density | Method | MMLU | HumanEval |
|---|---|---|---|
| | Single LLM | 0.8070 | 0.8171 |
| $\rho = 0.3$ | Vanilla MAS | 0.8140 | 0.8293 |
| | + MOC | 0.8351 | **0.8598** |
| | *Improv.* | ↑ 2.59% | ↑ 3.68% |
| $\rho = 0.5$ | Vanilla MAS | 0.8281 | 0.8476 |
| | + MOC | 0.8386 | 0.8537 |
| | *Improv.* | ↑ 1.27% | ↑ 0.72% |
| $\rho = 0.7$ | Vanilla MAS | 0.8211 | 0.8415 |
| | + MOC | **0.8526** | 0.8476 |
| | *Improv.* | ↑ 3.84% | ↑ 0.72% |

## 5.2. Main Results and Analysis

**Gains Across Backbones and Topologies.** Tables 1 and 2 demonstrate that MOC consistently outperforms Vanilla MAS across both Gemma-2-27B and Qwen2.5-32B backbones. While Vanilla MAS already improves over the single-agent setting through agent coordination, MOC further raises the performance ceiling under the same topologies. This suggests that the gains mainly come from the proposed communication scheme, including multi-order evidence exposure and structure-preserving consolidation, rather than model-specific tuning or topology optimization.

**Gemma-2-27B: Robust Improvements under Sparse Connectivity.** With Gemma-2-27B as the backbone, MOC improves the average performance across all edge densities, consistently outperforming Vanilla MAS from $\rho = 0.3$ to $\rho = 1.0$. The largest average improvement appears at $\rho = 0.3$, where MOC increases the average score from 0.7732 to 0.7910, corresponding to a relative gain of 2.30%. This trend suggests that MOC is particularly effective in sparse topologies, where direct neighbors provide limited evidence and multi-order message transmission can better expand the evidence receptive field. By exposing raw upstream responses from multiple hop orders, MOC preserves critical evidence that may be weakened during sequential transmission. This effect is especially evident on AQuA, where accuracy improves from 0.6969 to 0.7441 at $\rho = 0.3$. Although one slight degradation appears on SVAMP at $\rho = 1.0$, MOC still achieves a higher average score under the fully connected DAG, indicating robust overall gains across topological settings.

**Qwen2.5-32B: Scaling to Stronger Backbones.** On Qwen2.5-32B, MOC consistently outperforms Vanilla MAS across all evaluated edge densities on both MMLU and HumanEval. On MMLU, the largest improvement appears at $\rho = 0.7$, where the accuracy increases from 0.8211 to 0.8526, corresponding to a relative gain of 3.84%. On HumanEval, MOC achieves the best performance at $\rho = 0.3$, improving the Pass@1 score from 0.8293 to 0.8598 with a relative gain of 3.68%. These results suggest that multi-order message transmission remains beneficial on a stronger backbone, improving both general reasoning and code generation under the same topological settings.

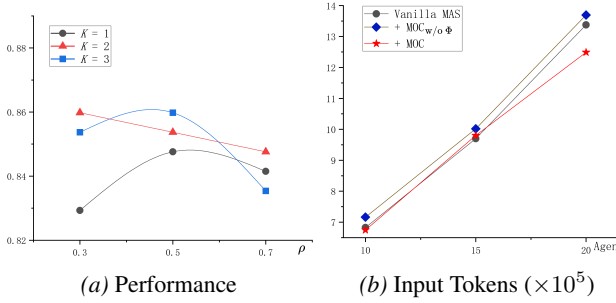

*(a)* Performance        *(b)* Input Tokens $(\times 10^5)$

*Figure 3.* Analysis of MOC on HumanEval with *Qwen2.5-32B-Instruct* based agents. (a) Performance across different orders $K$ under varying edge densities $\rho$. (b) Input token cost of MOC and variants under varying agent numbers.

### 5.3. Framework Analysis

**Effectiveness Analysis of Different Orders $K$.** Figure 3(a) shows that expanding the receptive field from $K = 1$ to $K = 2$ consistently improves performance across all tested edge densities. Specifically, the largest improvement appears at $\rho = 0.3$, where the accuracy increases from 0.8293 to 0.8598, corresponding to a relative gain of 3.68%. This suggests that exposing raw upstream evidence from higher-order dependencies helps alleviate semantic attenuation by providing broader contextual information. However, further increasing the order to $K = 3$ does not consistently bring additional gains. Although $K = 3$ achieves the best result of 0.8598 at $\rho = 0.5$, it drops to 0.8354 at $\rho = 0.7$, suggesting that excessive higher-order evidence may introduce redundancy in denser graphs. This non-monotonic trend indicates that a moderate receptive field is more stable, making $K = 2$ a robust default in our experiments.

**Ablation Study on Consolidation Operator $\Phi$.** Figure 3(b) compares the input token cost of Vanilla MAS, $\text{MOC}_{w/o\Phi}$, and MOC under different numbers of agents. Compared with $\text{MOC}_{w/o\Phi}$, MOC consistently reduces token consumption across all tested agent numbers, showing that the consolidation operator can effectively remove redundant multi-order evidence before context composition. Notably, MOC can even use fewer tokens than Vanilla MAS at 10 and 20 agents, because $\Phi$ enforces the topology-aware message-count budget $B_{\text{msg}}$ and applies length-controlled distillation to consolidated messages. At 20 agents, MOC decreases the input tokens from $13.69 \times 10^5$ to $12.49 \times 10^5$, demonstrating the necessity of $\Phi$ for controlling context growth in larger agent systems.

**Efficiency Analysis of Consolidation Operator $\Phi$.** Table 3 reports the runtime breakdown of MOC with seven agents on a single MMLU sample. The overhead is dominated by the distillation stage, while embedding computation and similarity estimation are negligible, taking only 0.25s at $\rho = 0.5$ and 0.31s at $\rho = 0.7$ in total. Since each merge

*Table 3.* Runtime breakdown (s) of MOC with seven *Gemma-2-27B-Instruct* based agents under different edge densities $\rho$.

| $\rho$ | Total | Embedding | Similarity | Distillation |
|---|---|---|---|---|
| 0.5 | 79.96 | 0.24 | 0.01 | 79.70 |
| 0.7 | 83.34 | 0.30 | 0.01 | 83.03 |

operation generates $M = 5$ merged candidates using the distillation model before selection, the main runtime cost comes from candidate generation during distillation.

### 5.4. Further Analysis

**MOC on Task-adaptive Topologies.** Beyond random DAG backbones, we integrate MOC with task-adaptive graphs produced by G-Designer (Zhang et al., 2025b). As shown in Fig. 4, MOC improves performance by 1.8% on MMLU, 1.0% on SVAMP, and 4.8% on HumanEval, demonstrating that our scheme is complementary to topology optimization. Regarding resource usage, the total input tokens of MAS increase by 14.1%, 11.1%, and 40.1% respectively, where these figures exclude consolidation costs. The larger increase on HumanEval likely comes from longer code-centric messages and the sparse dynamic topologies produced by G-Designer, where the multi-order evidence surfaced by MOC does not sufficiently trigger the consolidation operator $\Phi$. This highlights a natural trade-off between richer evidence exposure and context cost, though MOC remains effective even under specialized, dynamically crafted graphs.

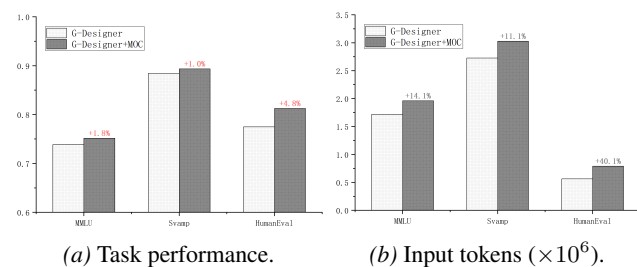

*(a)* Task performance.      *(b)* Input tokens $(\times 10^6)$.

*Figure 4.* Generalization study of MOC on task-adaptive G-Designer topologies using the *Gemma-2-27B-Instruct* based agents. (a) Task performance across MMLU, SVAMP, and HumanEval benchmarks. (b) Total MAS input token consumption.

**Generalization to Larger LLMs.** To evaluate model-agnosticism, we test MOC using DeepSeek-V3.2-685B-Chat based agents, where Table 4 shows improvements of 0.38% on MMLU and 0.82% on MMLU-Pro. Although input tokens increase by up to 22.8%, output tokens for MMLU-Pro drop from 795k to 759k, indicating that richer upstream evidence enables more decisive generation with less redundancy. These results confirm that MOC remains robust even when paired with state-of-the-art models under non-deterministic decoding.

*Table 4.* Generalization study on MOC using *DeepSeek-V3.2-685B-Chat* based agents across MMLU and MMLU-Pro benchmarks.

| Method | Acc. | In Tokens | Out Tokens |
|---|---|---|---|
| *MMLU* | | | |
| Single LLM | 0.8772 | 52,904 | 1,140 |
| Vanilla MAS | 0.9228 | 1,958,918 | 490,362 |
| + MOC | **0.9263** | 2,405,945 | 486,600 |
| *MMLU-Pro* | | | |
| Single LLM | 0.6679 | 85,260 | 1,120 |
| Vanilla MAS | 0.8821 | 3,178,655 | 794,530 |
| + MOC | **0.8893** | 3,722,630 | 759,219 |

## 6. Conclusion

In this paper, we addressed the critical yet underexplored problem of effective message transmission and optimization in LLM-based MAS. We proposed Multi-Order Communication (MOC), a topology-aware scheme that expands the evidence receptive field by surfacing raw upstream responses across multiple hop distances. To ensure practical efficiency, we introduced a Semantic-Topological Merging algorithm that prunes semantic redundancy while strictly honoring the communication hierarchy.

*Limitations and Future Works.* While MOC shows promising results in enhancing LLM-based MAS, its communication order $K$ is currently fixed rather than adaptively determined. Although higher-order evidence can enrich the target agent's context, it may also introduce incorrect reasoning or even potentially malicious messages from compromised agents. Future work will explore adaptive and trust-aware schemes that dynamically select both the hop range and upstream messages based on topology structure, evidence consistency, and message reliability, thereby improving robustness while preserving communication efficiency.

## Impact Statement

**Ethical considerations.** The proposed MOC framework is designed to advance communication scheme optimization in LLM-based multi-agent systems. It operates by reorganizing and consolidating existing agent outputs according to their topological dependencies. We affirm that MOC does not introduce additional ethical concerns regarding its motivation, design, implementation, or data usage.

**Societal implications.** MOC improves the communication scheme in LLM-based multi-agent systems by enhancing multi-hop information transmission and reducing redundant message exchange. This approach supports evidence-intensive and detail-sensitive tasks that require aggregation, reasoning, and verification. By making agent collaboration more coherent and resource-efficient, MOC has the potential to contribute to the practical development of LLM-based multi-agent systems.

## Acknowledgements

This work was supported by grants from the National Natural Science Foundation of China (72595845, 72595840) and the Yangtze River Delta Science and Technology Innovation Community Joint Research Project (YDZX20233100004031).

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

# A. GNN Message Passing Paradigm vs. LLM-based MAS Communication Scheme

We relate our inter-agent communication scheme to the standard Message Passing (MP) paradigm in Graph Neural Networks (GNNs), clarifying their structural alignment and fundamental differences.

## A.1. GNN Message Passing Paradigm

In the general message passing paradigm of GNNs, the update of a node $v_j$ is typically decomposed into neighborhood aggregation and node state update. Let $\mathbf{h}_i \in \mathbb{R}^d$ denote the feature vector of node $i$. A representative formulation can be written as:

$$\mathbf{m}_j = \text{AGG}\left(\{\mathbf{h}_i\}_{v_i \in \mathcal{N}_j^{\text{in}}}\right) = \sum_{v_i \in \mathcal{N}_j^{\text{in}}} \alpha_{ji}\mathbf{h}_i, \qquad \text{(neighbor message aggregation)}$$

$$\mathbf{h}_j' = \text{UPDATE}(\mathbf{h}_j, \mathbf{m}_j) = \sigma\left(\mathbf{W} \cdot \text{COMBINE}(\mathbf{h}_j, \mathbf{m}_j)\right), \quad \text{(node state update)}$$

(17)

where $\alpha_{ji}$ denotes the aggregation coefficient from node $v_i$ to node $v_j$, $\mathbf{W}$ represents learnable weights, and $\sigma(\cdot)$ denotes a non-linear activation function. For example, in GCN (Kipf & Welling, 2017), the aggregation weight $\alpha_{ji}$ is instantiated by the degree-normalized adjacency weight, and the self-loop allows the node's own representation to be combined with its neighborhood information. In mean-aggregation variants such as GraphSAGE (Hamilton et al., 2017), $\alpha_{ji}$ can be viewed as a uniform averaging weight over the sampled neighbors.

## A.2. LLM-based MAS Communication Scheme

As defined in Definition 3.1 and Definition 3.2, the communication logic for an agent $v_j$ in our framework follows a two-stage process operating in the space of discrete natural language:

$$\mathcal{C}_j = \Psi\left(\{\mathcal{R}_i\}_{v_i \in \mathcal{N}_j^{\text{in}}} \mid \pi_j\right) = \bigoplus_{k=1}^{n_j} \mathcal{R}_{\pi_j(k)}, \quad \text{(in-neighbor context composition)}$$

$$\mathcal{R}_j = f_j\left(\mathcal{P}_{sys,j} \oplus \mathcal{Q} \oplus \mathcal{C}_j\right), \qquad \text{(agent response generation)}$$

(18)

where $\mathcal{R}_i$ acts as a discrete "message", and the LLM generative function $f_j(\cdot)$ serves as a high-capacity, non-linear update operator.

## A.3. Key Differences

**Permutation Invariance vs. Order Sensitivity.** In Eq. (17), AGG is typically permutation-invariant (e.g., sum, mean) to treat neighbors as an unordered set. In Eq. (18), due to the auto-regressive nature of LLMs, the composition $\Psi$ is highly order-sensitive. The sequence $\pi_j$ imposes a deterministic message order, as the relative position of messages can affect the attention pattern and generated response.

**Feature Transformation vs. Semantic Inference.** While GNNs use linear projections and activations to refine continuous embeddings, LLM-based agents perform *semantic inference*. The operator $f_j$ does not merely aggregate features but synthesizes evidence, filters noise, and generates task-specific responses conditioned on the context $\mathcal{C}_j$.

**Fixed-dimensional vs. Variable-length.** GNN aggregation maps neighbors to a fixed-dimensional vector $\mathbf{m}_j \in \mathbb{R}^d$. In contrast, MAS aggregation results in variable-length text $\mathcal{C}_j$. As the number of in-neighbors $n_j$ grows, the token length of $\mathcal{C}_j$ increases, which introduces constraints related to the LLM's context window and computational budget.

# B. Experimental Setups

## B.1. Datasets Details

To evaluate the effectiveness of **MOC** across diverse tasks while maintaining computational efficiency, we curate specific test subsets from six benchmarks covering general reasoning, mathematics, and code generation. Table 5 provides the detailed statistics, metrics, sampling methods, and license information.

**Data Selection and Sources.** We utilize the following specific repositories and sampling methods: (1) **MMLU** (`cais/mmlu`) (Hendrycks et al., 2021): A subset of 5 randomly sampled questions from each of the 57 test categories.

(2) **MMLU-Pro** (`TIGER-Lab/MMLU-Pro`) (Wang et al., 2024): 20 samples randomly selected from each of the 14 consolidated categories. (3) **GSM8K** (`openai/gsm8k`) (Cobbe et al., 2021): A random sample of 300 instances from the official test set. (4) **SVAMP** (`ChilleD/SVAMP`) (Patel et al., 2021): The complete test set of 300 instances. (5) **AQuA** (`deepmind/aqua_rat`) (Ling et al., 2017): The standard test set containing 254 algebraic problems. (6) **HumanEval** (`openai/openai_humaneval`) (Chen et al., 2021): The full set of 164 Python programming tasks.

**Evaluation Protocol.** For all benchmarks except HumanEval, we utilize exact match accuracy (Acc.) of the predicted option or numerical value. For HumanEval, we report the *Pass@1* metric calculated via functional unit tests with greedy decoding.

*Table 5.* Summary of dataset statistics, sampling strategies, and license information.

| Category | Dataset | Answer Type | Metric | Sampling Strategy | #Samples | License |
|---|---|---|---|---|---|---|
| General reasoning | MMLU | Four-choices | Acc. | 5 per cat (57 cats) | 285 | MIT License |
| | MMLU-Pro | Ten-choices | Acc. | 20 per cat (14 cats) | 280 | CDLA-Permissive-2.0 |
| Math reasoning | GSM8K | Number | Acc. | Random subset | 300 | MIT License |
| | SVAMP | Number | Acc. | Full Test Set | 300 | MIT License |
| | AQuA | Five-choices | Acc. | Full Test Set | 254 | Apache-2.0 |
| Code generation | HumanEval | Code | Pass@1 | Full Test Set | 164 | MIT License |

## B.2. Random DAG topology generation

For synthetic communication graphs with $N$ agents, we construct a random directed acyclic graph (DAG) by first sampling a random topological order $\sigma$, i.e., a uniform shuffle of $\{0, \ldots, N-1\}$, and then only allowing *forward* edges consistent with $\sigma$. Let $\text{pos}(\cdot)$ denote the position of a node in $\sigma$. The candidate set of acyclic edges is

$$E_{\text{cand}} = \{(i,j) \mid i \neq j, \ \text{pos}(i) < \text{pos}(j)\}, \qquad |E_{\text{cand}}| = \frac{N(N-1)}{2}. \tag{19}$$

To avoid isolated agents, i.e., nodes with both zero in-degree and zero out-degree, we first add a chain backbone along $\sigma$:

$$E_{\text{chain}} = \{(\sigma_k, \sigma_{k+1})\}_{k=1}^{N-1}, \tag{20}$$

which is acyclic and covers all nodes. This non-isolation constraint requires at least $N-1$ edges, while the complete DAG under $\sigma$ contains at most $|E_{\text{cand}}|$ edges.

Given an edge density $\rho \in (0, 1]$, we set the target number of edges as

$$m = \lceil \rho \cdot |E_{\text{cand}}| \rceil, \tag{21}$$

with $m$ restricted to $[N-1, |E_{\text{cand}}|]$, where the lower bound ensures that all nodes are covered and the upper bound corresponds to the complete DAG under $\sigma$. After adding $E_{\text{chain}}$, we uniformly sample without replacement the remaining $m - |E_{\text{chain}}|$ edges from $E_{\text{cand}} \setminus E_{\text{chain}}$ and add them to form the final edge set $E$.

## B.3. Detailed Merging Prompts

```
Prompt Variant 1: Narrative Synthesis

Synthesize the messages from AGENT_1 and AGENT_2 into a single, cohesive update.
Target length:  approximately {κ}% of the original token count.
Task:  Merge overlapping information and deduplicate common findings while
preserving both agents' distinct contributions.
Constraint:  Do not add new information.  Output ONLY the synthesized text with no
preamble.

AGENT_1:  [ID:{ IDᵢ} | Role:{Roleᵢ} ]
{Contentᵢ}
AGENT_2:  [ID:{ IDⱼ} | Role:{Roleⱼ} ]
{Contentⱼ}
```

**Prompt Variant 2: Logical Integrity**

```
Merge the communications from AGENT_1 and AGENT_2.
Target length: roughly {κ}% of the original volume.
Task: Compress the text but strictly retain the complete reasoning chain and all
logical dependencies leading to the conclusion.
Constraint: Do not add new information. Output ONLY the synthesized text with no
preamble.

AGENT_1: [ID: {ID_i} | Role: {Role_i}]
{Content_i}
AGENT_2: [ID: {ID_j} | Role: {Role_j}]
{Content_j}
```

**Prompt Variant 3: Technical Precision**

```
Consolidate the data from AGENT_1 and AGENT_2 into a high-density summary.
Target length: around {κ}% of original tokens.
Task: Ensure zero-loss for all Agent IDs, technical parameters, formulas, and
specific values. Strip away all conversational fillers.
Constraint: Do not add new information. Output ONLY the synthesized text with no
preamble.

AGENT_1: [ID: {ID_i} | Role: {Role_i}]
{Content_i}
AGENT_2: [ID: {ID_j} | Role: {Role_j}]
{Content_j}
```

**Prompt Variant 4: Actionable Intelligence**

```
Combine AGENT_1 and AGENT_2 messages into a "telegram-style" actionable update.
Target length: approximately {κ}% volume.
Task: Prioritize actionable data and final decisions. Use shorthand where possible
while maintaining source attribution for key facts.
Constraint: Do not add new information. Output ONLY the synthesized text with no
preamble.

AGENT_1: [ID: {ID_i} | Role: {Role_i}]
{Content_i}
AGENT_2: [ID: {ID_j} | Role: {Role_j}]
{Content_j}
```

**Prompt Variant 5: Deduplication & Structure**

```
Integrate the content from AGENT_1 and AGENT_2 while maintaining any structural
headers.
Target length: about {κ}% of the original count.
Task: Identify and merge redundant statements between the two agents to maximize
information density per token.
Constraint: Do not add new information. Output ONLY the synthesized text with no
preamble.

AGENT_1: [ID: {ID_i} | Role: {Role_i}]
{Content_i}
AGENT_2: [ID: {ID_j} | Role: {Role_j}]
{Content_j}
```

## C. Visualization of Graph Structure

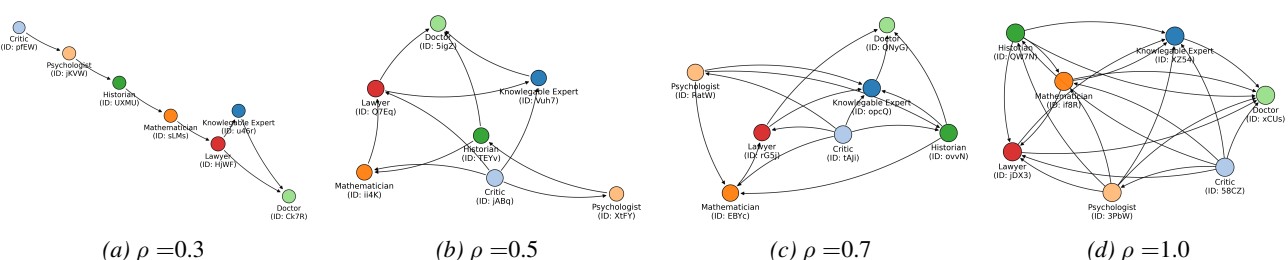

(a) $\rho = 0.3$     (b) $\rho = 0.5$     (c) $\rho = 0.7$     (d) $\rho = 1.0$

*Figure 5.* Examples of communication graph structures generated on the MMLU dataset under different edge density $\rho$.

## D. Case Study

**Original #1: Role: Critic**

**Message:**
Agent vpjG, role is Critic, output is: Let's analyze each statement:
Statement 1: An abelian group of order 45 can be decomposed into its prime factors: $3^2 \times 5$. By Lagrange's Theorem, the order of a subgroup must divide the order of the group. Therefore, it's possible to have subgroups of orders 1, 3, 5, 9, 15, and 45. A subgroup of order 10 is not possible. So, Statement 1 is False.
Statement 2: This statement accurately describes the definition of a normal subgroup. A subgroup $H$ is normal in $G$ if and only if $gHg^{-1} = H$ for all $g \in G$. This condition implies that the left and right cosets of $H$ coincide. So, Statement 2 is True.
The answer is B

**Merged: Role: Critic & Historian**

**Message:**
Statement 1: An abelian group of order 45 can be decomposed into its prime factors: $3^2 \times 5$. By Lagrange's Theorem, the order of a subgroup must divide the order of the group. Therefore, it's possible to have subgroups of orders 1, 3, 5, 9, 15, and 45. A subgroup of order 10 is not possible. So, Statement 1 is False.
Statement 2: A subgroup $H$ is normal in $G$ if and only if $gHg^{-1} = H$ for all $g \in G$. This condition implies that the left and right cosets of $H$ coincide. So, Statement 2 is True. The answer is B

**Original #2: Role: Historian**

**Message:**
Agent 6xho, role is Historian, output is: Let's examine each statement:
Statement 1: An abelian group of order 45 has possible subgroup orders that divide 45 ($3^2 \times 5$). These are 1, 3, 5, 9, 15, and 45. A subgroup of order 10 is not possible. Therefore, Statement 1 is False.
Statement 2: The definition of a normal subgroup states that for every element $g$ in the group $G$, $gHg^{-1} = H$. This means left and right cosets coincide, making the number of left cosets equal to the number of right cosets. So, Statement 2 is True.
The answer is B

*Figure 6.* An example on the MMLU dataset of merging two agent messages with high semantic similarity, where the left side shows the original messages and the right side shows the merged message.

# E. Additional Results

*Table 6.* Detailed evaluation on HumanEval across various edge densities $\rho$ and communication hops $K$. **Agent Tokens** represent the prompt and completion overhead incurred by target multi-agents system. **Compressed Tokens** denote the prompt and completion overhead incurred by the operator $\Phi$. Bold values indicate the highest accuracy achieved within each density group.

| $\rho$ | $K$ | Acc. | Agent Tokens | | | Compressed Tokens | | |
|---|---|---|---|---|---|---|---|---|
| | | | Prompt | Compl. | Total | Prompt | Compl. | Total |
| | 1 | 0.8293 | 594,629 | 133,708 | 728,337 | 0 | 0 | 0 |
| 0.3 | 2 | **0.8598** | 654,313 | 126,526 | 780,839 | 0 | 0 | 0 |
| | 3 | 0.8537 | 680,157 | 124,636 | 804,793 | 0 | 0 | 0 |
| | 1 | 0.8476 | 614,974 | 130,277 | 745,251 | 0 | 0 | 0 |
| 0.5 | 2 | 0.8537 | 675,732 | 121,460 | 797,192 | 255,840 | 80,247 | 336,087 |
| | 3 | **0.8598** | 688,977 | 123,536 | 812,513 | 653,320 | 217,304 | 870,624 |
| | 1 | 0.8415 | 665,990 | 140,929 | 806,919 | 0 | 0 | 0 |
| 0.7 | 2 | **0.8476** | 685,458 | 120,247 | 805,705 | 627,250 | 209,466 | 836,716 |
| | 3 | 0.8354 | 699,396 | 126,490 | 825,886 | 649,263 | 216,566 | 865,829 |

