# OpenReview forum: "MOC: Multi-Order Communication in LLM-based Multi-Agent Systems"
_ICML.cc/2026/Conference — ICML 2026 regular_

### Official Review · Reviewer_fnhx · 2026-03-12

**Soundness:** 3
**Presentation:** 2
**Significance:** 2
**Originality:** 3
**Overall Recommendation:** 4
**Confidence:** 3

**Summary:**

MOC addresses the communication scheme design problem in LLM-based multi-agent systems by allowing target agents to directly access raw upstream responses from multiple hop distances (K-hop) instead of relying only on immediate neighbors. It introduces a Semantic-Topological Merging algorithm to compress redundant messages while preserving topological order, achieving consistent performance improvements across six benchmarks while stabilizing token usage.

**Compliance With Llm Reviewing Policy:**

Affirmed.

**Final Justification:**

Thank you for your detailed reply. I have updated my final justification.

**Key Questions For Authors:**

- It would be helpful for the authors to report the number of agents in the graph and to analyze how performance scales with an increasing number of agents.
- The computational cost of the model used for context compression should be explicitly considered in the evaluation.
- It is unclear how errors made by earlier agents are handled. In multi-agent settings, such errors might be recognized and corrected by the following agents, but when transferring two-hop context, there is a risk that these errors could propagate or even amplify.

**Limitations:**

see key questions

**Strengths And Weaknesses:**

strength：The paper identifies a critical yet overlooked research problem and provides rigorous graph-based formalization of intra-round multi-agent communication. Extensive experiments demonstrate consistent gains across diverse datasets and multiple model backbones. The proposed Semantic-Topological Merging effectively solves the context explosion problem, reducing token consumption.

weaknesses：

- The paper lacks a comparison with a baseline that simply collects context from 1-hop and 2-hop neighbors and concatenates it directly, which would help isolate the benefit of the proposed multi-hop communication mechanism. And experiments are mainly conducted on simple reasoning benchmarks.

---

> ### Author Rebuttal · Authors · 2026-03-31
>
> We sincerely thank you for the constructive comments and suggestions. We respond below to the questions on baselines, scalability, computational cost, and error propagation.
>
> > `W1a:` Comparison with direct concatenation of 1-hop and 2-hop contexts.
>
> Thanks for highlighting the importance of additional baselines to better isolate the contribution of MOC. Table A compares four settings on MMLU and HumanEval: (1) Vanilla MAS; (2) Naive 1-hop + 2-hop concatenation, which directly concatenates 1-hop and 2-hop neighbor contexts; (3) Single-hop with more edges, which adds shortcut edges to increase single-hop connectivity; (4) MOC.
>
> *Table A. Performance comparison on MMLU and HumanEval datasets.*
>
> | Method | MMLU | HumanEval |
> |---|---:|---:|
> | Vanilla MAS | 0.7789 | 0.7744 |
> | Naive 1-hop + 2-hop concatenation | 0.7789 | **0.7866** |
> | Single-hop with more edges | 0.7754 | 0.7378 |
> | **MOC** | **0.7895** | **0.7866** |
>
> Table A shows that Naive 1-hop + 2-hop concatenation does not improve over Vanilla MAS on MMLU, but improves HumanEval and matches MOC. This suggests that broader evidence exposure can help in some cases, but not consistently across tasks. Further evidence comes from the Single-hop with more edges baseline, which performs worse than both Vanilla MAS and MOC on both benchmarks, indicating that the gains of MOC cannot be explained simply by increasing single-hop connectivity. Instead, MOC works with the original communication topology by exposing multi-hop upstream evidence and consolidating it in a structured, topology-aware manner under a context budget. Therefore, its advantage lies not only in broader evidence access, but also in structured organization and compression. We will clarify this distinction and include the additional comparisons in the revised manuscript.
>
> ---
>
> > `W1b:` Evaluation on simple reasoning benchmarks.
>
> We would like to clarify that our evaluation is not limited to simple reasoning benchmarks. It covers six benchmarks spanning general reasoning, mathematical reasoning, and code generation, including more challenging settings such as MMLU-Pro and HumanEval.
>
> ---
>
> > `Q1:` Number of agents and scaling behavior of MOC.
>
> Our experiments use **7-agent** MAS communication graphs, which we will clarify in the revised manuscript. Table B reports results of Vanilla MAS and MOC with 5, 7, and 9 agents on MMLU. MOC consistently outperforms Vanilla MAS across all tested settings, suggesting that its advantage is stable across different agent counts.
>
> *Table B. Performance with various numbers of agents on MMLU dataset.*
>
> | Agent num | MAS | MOC |
> |---|---:|---:|
> | 5 | 0.7614 | **0.7930** |
> | 7 | 0.7789 | **0.7895** |
> | 9 | 0.7860 | **0.7930** |
>
> ---
>
> > `Q2:` What is the computational cost of context compression?
>
> To address this point, we report the average runtime and token consumption per sample of Vanilla MAS and MOC on MMLU in Table C.
>
> *Table C. Average computational cost per sample on MMLU dataset.*
>
> | Method | Time (s) | System Tokens | Compression Tokens |
> |---|---:|---:|---:|
> | Vanilla MAS | 21.47 | 5,342 | 0 |
> | MOC | 30.80 | 5,870 | 2,772 |
>
> Compared with Vanilla MAS, MOC increases runtime from 21.47s to 30.80s per sample, increases system tokens from 5,342 to 5,870, and introduces 2,772 compression tokens per sample. This makes the performance-cost trade-off explicit. We will include this analysis in the revised manuscript.
>
> ---
>
> > `Q3:` How does MOC handle error propagation from earlier agents?
>
> We agree that error propagation is closely related to communication scheme design in multi-agent systems. In this work, we focus on one specific aspect of this problem, namely how to expose, organize, and compress multi-hop evidence within a communication graph. A direct treatment of error propagation—such as explicitly tracing, detecting, or correcting erroneous intermediate messages—is therefore not the main target of the current paper, although it remains an important direction for future work.
>
> As a complementary analysis, Table D reports HumanEval performance under different edge densities $\rho$ and hop distances $K$. Using 2-hop context consistently outperforms  $K=1$ (i.e., Vanilla MAS)  across all tested edge densities. This suggests that, in the 2-hop setting, upstream errors do not appear to have a substantial negative effect on downstream performance. By contrast, slight degradation appears only when the hop distance is further increased, suggesting that potential error-accumulation effects may become more relevant at longer hop distances.
>
> *Table D. Performance across edge densities $\rho$ and hop distances $K$ on HumanEval dataset.*
>
> | $\rho$ | $K=1$ | $K=2$ | $K=3$ |
> |---|---:|---:|---:|
> | 0.3 | 0.8293 | **0.8598** | 0.8537 |
> | 0.5 | 0.8476 | 0.8537 | **0.8598** |
> | 0.7 | 0.8415 | **0.8476** | 0.8354 |
>
> ---
> Overall, we hope that our responses can fully address your concerns and will be grateful for any feedback.

---

> > ### Author Rebuttal · Reviewer_fnhx · 2026-04-01
> >
> > My questions are fully resolved, and I update my recommendation accordingly.

---

> > > ### Author Response · Authors · 2026-04-06
> > >
> > > **Dear Reviewer fnhx,**
> > >
> > > Thank you very much for your constructive feedback and support of our work. Your comments helped us refine the presentation, strengthen the experimental validation, and improve the manuscript. We particularly appreciate your recognition that our work focuses on an important problem in LLM-based MAS, provides a graph-based formalization of intra-round communication, and demonstrates empirical gains with effective context compression.
> > >
> > > Thank you again for your time, thoughtful evaluation, and valuable support.
> > >
> > > Best regards,
> > >
> > > The Authors

---

### Official Review · Reviewer_XTMY · 2026-03-13

**Soundness:** 3
**Presentation:** 3
**Significance:** 4
**Originality:** 4
**Overall Recommendation:** 4
**Confidence:** 4

**Summary:**

The paper addresses the communication bottleneck in LLM-based Multi-Agent Systems. Current MAS typically pass messages only from immediate predecessors, which leads to semantic attenuation over multi-hop paths and restricts the evidence receptive field. To overcome this, the paper propose **Multi-Order Communication (MOC)**, a framework that exposes target agents to raw responses from multi-hop ancestors. To prevent the inevitable token length explosion associated with expanded evidence, they introduce a Semantic-Topological Merging algorithm. Extensive experiments across six benchmarks, varying edge densities, and multiple LLM backbones demonstrate that MOC improves reasoning performance while maintaining a strictly controlled context token budget.

**Compliance With Llm Reviewing Policy:**

Affirmed.

**Final Justification:**

Concerns are partially addressed, but remaining questions on scalability and robustness lead me to keep my original score (I already gave positive score).

**Key Questions For Authors:**

1. Can the authors provide a latency analysis comparing the Vanilla MAS pipeline against the MOC pipeline? the overhead introduced by the Semantic-Topological Merging loop during inference would be interesting to see.

2. How sensitive is the overall performance to the similarity threshold / token distillation ratio? Do these parameters require dataset-specific tuning, or does a single configuration generalize optimally across all evaluated tasks?

3. Did the authors experiment with making the multi-order hop parameter dynamic, perhaps adapting it based on an agent's specific in-degree or the complexity of the query?

**Limitations:**

Yes

**Strengths And Weaknesses:**

**Strengths**

1. By formulating a discrete analogue to GNN message passing and addressing the unique text-based bottlenecks of LLMs, the authors provide a highly practical and novel perspective on multi-agent collaboration.

2. The proposed MOC framework is well-founded. The mathematical formulation effectively translates graph reachability and topological sorting into structured prompt contexts.

3. The experimental validation is robust. The authors evaluate their framework across general reasoning, math, and code generation, demonstrating consistent gains over vanilla MAS baselines.

**Weaknesses**

1. The paper trys to present the key issue of token inefficiency, but it ignores the computational latency overhead introduced by the proposed solution. Executing an embedding model, computing similarity matrices, and running an auxiliary 9B distillation model iteratively at each communication node will inevitably introduce significant wall-clock latency.

2. The framework introduces several fixed hyperparameters, notably the similarity threshold, the context length ratio limit, and the hop parameter.

---

> ### Author Rebuttal · Authors · 2026-03-31
>
> Sincere thanks for the thoughtful and constructive comments on our manuscript. We appreciate your positive assessment of our motivation, formulation, and experimental validation. Below, we have carefully prepared a point-by-point reply.
>
>
> > `W1&Q1:` Latency Overhead of MOC Compared with Vanilla MAS.
>
> We thank you for raising the important question of inference latency. We agree that MOC introduces additional wall-clock overhead beyond Vanilla MAS, since it includes embedding-based similarity computation and semantic-topological consolidation before final context composition. To make this trade-off explicit, we have added a latency analysis on MMLU and HumanEval, as shown in Table A.
>
> *Table A. Inference time on MMLU and HumanEval datasets.*
>
> | Dataset | Method | Total Time (s) | Avg./sample (s) |
> |---|---|---:|---:|
> | MMLU | Vanilla MAS | 6119.689 | 21.47 |
> | MMLU | MOC | 8778.908 | 30.80 |
> | HumanEval | Vanilla MAS | 4773.562 | 29.11 |
> | HumanEval | MOC | 6103.027 | 37.21 |
>
> As expected, MOC introduces additional inference latency compared with Vanilla MAS. On MMLU, the average latency increases from 21.47s to 30.80s per sample, and on HumanEval, from 29.11s to 37.21s per sample. This overhead mainly comes from the semantic-topological merging procedure, including similarity-based message consolidation and distillation. We note that the current results are obtained under our default setting, where the similarity threshold is set to 0.01 and the distillation step uses 5 samples. In practice, the runtime overhead may be reduced by using a stricter similarity threshold or fewer samples, which would reduce the number of candidate merges and the corresponding distillation cost. We will include this latency analysis and a corresponding discussion in the revised manuscript. Exploring the trade-off between runtime and performance under different sampling and merging configurations is an important direction for future work.
>
> ---
>
> > `W2&Q2:` Sensitivity to similarity threshold and distillation ratio.
>
> We thank you for this important question. In our implementation, the batch-wise $\epsilon$-approximate merging threshold was fixed to $\epsilon = 0.01$, while the distillation ratio was fixed to $\kappa = 0.45$ across all experiments. We did not perform any dataset-specific tuning for these two parameters. This shared configuration across all experiments suggests that the observed gains mainly come from the MOC framework itself, rather than from careful hyperparameter optimization for individual datasets. At the same time, these parameters act as global control knobs for consolidation strength and influence the compression-fidelity trade-off during merging. We will revise the paper to clarify this point more explicitly and include additional sensitivity experiments in the revised version.
>
> ---
>
> > `Q3:` Did the authors consider a dynamic multi-order hop parameter?
>
> We thank you for this insightful suggestion. In the current work, we do not explicitly study a dynamic multi-order hop parameter, and instead adopt a fixed hop order in our experiments. Our current design aims to evaluate the effectiveness of MOC under a simple and controlled setting, so that the observed improvements can be more directly attributed to the framework itself. At the same time, a dynamic hop parameter is a meaningful extension, since different graph structures or tasks may benefit from different receptive fields. We will revise the paper to clarify this point and include a discussion of dynamic hop selection as an important direction for future work.
>
> ---
>
> We hope the above clarifications address your concerns and improve the presentation in the revised manuscript.

---

> > ### Author Rebuttal · Reviewer_XTMY · 2026-04-04
> >
> > Thank you for the rebuttal response. I have follow-up questions:
> >
> > 1. Could the authors provide a more detailed breakdown of where the additional time is spent (e.g., embedding, similarity computation, distillation), and how this overhead scales with graph size or number of agents?
> >
> > 2. Could the authors include quantitative sensitivity results to better understand the stability of MOC under different configurations?
> >
> > 3. Do the authors have any preliminary evidence or intuition on when increasing hop order begins to hurt performance, especially in denser graphs?
> >
> > In this regard, I will keep my current score.

---

> > > ### Author Response · Authors · 2026-04-06
> > >
> > > Thanks for the reply. Below we address the three follow-up points in turn.
> > >
> > > ---
> > >
> > > > `Q1:` Additional Runtime Analysis of MOC.
> > >
> > > Table A reports a detailed runtime breakdown of MOC overhead on a single sample under different agent numbers and graph densities. The results show that the additional overhead is dominated by the distillation stage, while embedding and similarity computation contribute only a very small portion. The overall cost also increases substantially with the number of agents, and can vary with graph density due to differences in candidate merge structure. We will add this breakdown to the revision to make the runtime trade-off more explicit.
> > >
> > >
> > > *Table A. Detailed breakdown on a single sample of MMLU dataset.*
> > >
> > > | Agent num | $\rho$ | Total (s) | Embedding (s) | Similarity (s) | Distillation (s) |
> > > |---|---|---:|---:|---:|---:|
> > > | 7 | 0.5 | 79.9578 | 0.2434 | 0.0110 | 79.7034 |
> > > | 7 | 0.7 | 83.3435 | 0.3016 | 0.0120 | 83.0299 |
> > > | 9 | 0.5 | 331.7181 | 0.8166 | 0.0410 | 330.8605 |
> > > | 9 | 0.7 | 304.0569 | 1.0251 | 0.0394 | 302.9924 |
> > >
> > > ---
> > >
> > > > `Q2:` Additional Sensitivity Analysis of Key Hyperparameters.
> > >
> > > Table B reports a compact sensitivity check over the merging threshold $\epsilon$ and distillation ratio $\kappa$. The accuracy varies only slightly across these settings, which suggests that MOC is not overly sensitive to moderate changes in these two hyperparameters. This indicates that the gains are not tied to a narrowly tuned configuration.
> > >
> > > *Table B. Performance under different values of $\epsilon$ and $\kappa$ on MMLU dataset.*
> > >
> > > | $\epsilon$ $\backslash$ $\kappa$ | 0.45 | 0.55 |
> > > |---|---:|---:|
> > > | 0.01 | 0.7895 | 0.7825 |
> > > | 0.05 | 0.7825 | 0.7860 |
> > >
> > > ---
> > >
> > > > `Q3:` Discussion on the Hop Order.
> > >
> > > Table C reports performance under different edge densities $\rho$ and hop distances $K$ on HumanEval dataset. Our existing results suggest that, relative to the baseline setting ($K=1$), $K=2$ consistently improves performance, while the effect of $K=3$ depends on graph density. In Table C, $K=3$ remains effective at lower or moderate densities, but begins to degrade under denser graphs (e.g., $\rho=0.7$). Our intuition is that, in denser graphs, increasing hop order exposes each agent to more overlapping and indirectly relayed messages, so the marginal gain from additional evidence decreases while redundancy accumulates more quickly. This suggests that the issue is not multi-hop communication itself, but the increased redundancy brought by larger hop orders in dense graphs.
> > >
> > > *Table C. Performance across edge densities $\rho$ and hop distances $K$ on HumanEval dataset.*
> > >
> > > | $\rho$ | Vanilla MAS ($K=1$) | $K=2$ | $K=3$ |
> > > |---|---:|---:|---:|
> > > | 0.3 | 0.8293 | **0.8598** | 0.8537 |
> > > | 0.5 | 0.8476 | 0.8537 | **0.8598** |
> > > | 0.7 | 0.8415 | **0.8476** | 0.8354 |

---

### Official Review · Reviewer_4ap9 · 2026-03-14

**Soundness:** 2
**Presentation:** 3
**Significance:** 2
**Originality:** 2
**Overall Recommendation:** 3
**Confidence:** 3

**Summary:**

MOC is a topology-aware communication scheme that improves multi-agent system performance by surfacing raw multi-hop upstream evidence and employing a semantic-topological merging algorithm to optimize context efficiency.

**Compliance With Llm Reviewing Policy:**

Affirmed.

**Final Justification:**

I believe the original paper should have included the most important comparative experiments and a discussion of related work, which would have made the paper more complete and convincing.

**Key Questions For Authors:**

Could you provide more evidence to justify the **necessity** of introducing multi-hop communication instead of substituting it with equivalent traditional single-hop methods?

**Limitations:**

No. Please discuss the application scenarios of the multi-hop reasoning design. As noted in the Weakness section, the necessity of introducing this design should be clearly justified.

**Strengths And Weaknesses:**

Strengths:

Experiments are conducted on both static and dynamic workflows, and performance improvements are demonstrated.

Weaknesses:

1. The paper missed a **critical comparison** between the performance of multi-hop message passing and single-hop with more edges. For example, what is the performance difference if we treat a two-hop communication on $A \to B \to C$ as three single-hop communications ($A \to B$, $B \to C$, and $A \to C$)? Such a comparison is needed to justify the **necessity** of introducing multi-hop communication.
2. Some traditional MAS algorithms could also be considered (or at least discussed) to model the connection strength between each pair of agents, e.g., [1910.00091] *Deep Coordination Graphs*; [2211.08404] *Non-Linear Coordination Graphs*. Although their message-passing protocols directly serve to compute payoffs, similar ideas could be considered to model message passing among LLM agents.

---

> ### Author Rebuttal · Authors · 2026-03-31
>
> We would like to express our sincere respect for your careful and constructive comments. We particularly appreciate your emphasis on clarifying the necessity of multi-hop communication and on positioning our method against relevant alternatives. Below, we provide a point-by-point response to address these concerns.
>
> > `W1&Q1:` Comparison between multi-hop message passing and single-hop with more edges.
>
> We thank you for pointing out the importance of comparing multi-hop message passing with single-hop communication augmented by additional edges. We clarify that multi-hop communication is necessary because simply replacing multi-hop dependencies with additional single-hop edges is not an equivalent substitute. Adding edges changes the graph topology and execution dependencies, whereas MOC preserves the original DAG and execution order, and only expands the evidence accessible to the target agent within a communication round. More importantly, MOC organizes upstream evidence according to both hop order and topological order, allowing the target agent to access raw multi-hop sources and verify intermediate restatements, rather than relying only on first-order rewritten summaries. Therefore, MOC is not merely “letting A’s message reach C,” but a structured multi-hop communication scheme designed to preserve provenance and improve evidence utilization. In Table A, we present the performance of multi-hop message passing and single-hop with more edges on MMLU and HumanEval datasets.
>
> *Table A. Performance comparison of Vanilla MAS, single-hop with more edges and MOC on MMLU and HumanEval datasets.*
>
> | Method                      | MMLU    | HumanEval |
> |------------------------------|---------|-----------|
> | Vanilla MAS                  | 0.7789  | 0.7744    |
> | single-hop with more edges   | 0.7754  | 0.7378    |
> | **MOC**                     | **0.7895**  | **0.7866** |
>
> The results show that simply adding single-hop connections does not guarantee improvement, and can even degrade performance, as observed on HumanEval. This suggests that naive edge augmentation may disrupt an otherwise effective communication topology. In contrast, MOC achieves consistent gains while preserving the original execution structure. These results indicate that the benefit of MOC does not come from “more connectivity” alone, but from structured multi-hop evidence exposure and organization. We will clarify this distinction and include the additional comparison in the revised manuscript.
>
> ---
>
> > `W2:` Discussion of Traditional MAS algorithms.
>
> We appreciate your suggestion regarding traditional MAS algorithms. Works such as *Deep Coordination Graphs* and *Non-Linear Coordination Graphs* are **related and potentially informative**, as they were originally developed for numeric payoff optimization, and their graph-based message-passing ideas could inspire modeling multi-hop communication among LLM agents. At the same time, MOC specifically addresses **LLM agents’ multi-hop communication**, which is order-sensitive, semantic-inference based, and context-budgeted. Incorporating concepts such as relation-strength modeling or adaptive weighting from traditional MAS into MOC is a promising direction for future work, and we would be happy to include a discussion of this perspective in the revised manuscript.
>
> ---
>
> > `Limitations:` Discussion of MOC's application scenarios.
>
> Thanks for the suggestion. We agree that a discussion of MOC’s applicability scenarios is important. Based on our experiments and analysis, MOC is especially beneficial in the following settings:
>
> 1. **Sparse or fixed-topology communication graphs**, where multi-hop exposure can compensate for limited direct connectivity and improve communication beyond what task-adaptive topology alone may capture.
> 2. **Source verification–sensitive tasks**, where intermediate agent restatements need to be validated against upstream evidence.
> 3. **Code or detail-sensitive tasks**, where retaining fine-grained information across multiple hops is critical.
>
> At the same time, MOC is not universally advantageous. When the communication graph is already dense, or when extra long-range evidence becomes redundant, the marginal benefit may diminish, and larger hop orders may even hurt performance in some cases. We will include this discussion in the revision to clarify MOC’s strengths and limitations.
>
> ---
>
> Overall, we hope that our responses can fully address your concerns and will be grateful for any feedback.

---

> > ### Author Rebuttal · Reviewer_4ap9 · 2026-04-03
> >
> > I think the authors have addressed most of my concerns. Incorporating these discussions into the paper would further strengthen it.

---

> > > ### Author Response · Authors · 2026-04-06
> > >
> > > **Dear Reviewer 4ap9,**
> > >
> > > Thank you for your careful review and constructive feedback. We truly appreciate your comments on the comparison with single-hop communication augmented by additional edges, the discussion of traditional MAS algorithms, and the application scenarios of our method.
> > >
> > > We will incorporate these discussions and corresponding clarifications into the final manuscript to further strengthen the paper.
> > >
> > > Best regards,
> > >
> > > The Authors

---

### Decision · Program_Chairs · 2026-04-30

**Decision:**

Accept (regular)

**Comment:**

This paper addresses a highly relevant bottleneck in LLM-based Multi-Agent Systems by proposing the Multi-Order Communication framework, which thoughtfully leverages multi-hop evidence streams and semantic-topological merging to improve inter-agent message passing. Reviewers commended the theoretical formalization of the problem and the promising performance gains demonstrated across multiple reasoning and coding benchmarks. However, the experimental evaluation remains somewhat limited; the framework was primarily tested on small-scale configurations, lacks sufficient baseline comparisons, and reviewers raised valid concerns regarding the substantial latency overhead introduced by the distillation step. Despite these limitations in the experimental scope, the authors' rebuttal satisfactorily addressed the missing baseline comparisons, and the conceptual novelty of structured multi-hop communication offers a valuable contribution to the field.